# Association of enterolactone with blood pressure and hypertension risk in NHANES

**Cynthia M. Weiner**[1], **Shannon E. Khan**[1], **Caleb Leong**[2], **Sushant M. Ranadive**[1], **Sara C. Campbell**[3], **Jeffrey T. Howard**[2], **Kevin S. Heffernan**[4]*

1 Department of Kinesiology, University of Maryland, College Park, Maryland, United States of America,
2 Department of Public Health, University of Texas at San Antonio, One UTSA Circle, San Antonio, Texas, United States of America, 3 Department of Kinesiology and Health, Rutgers University, New Brunswick, New Jersey, United States of America, 4 Department of Exercise Science, Syracuse University, Syracuse, NY, United States of America

* ksheffer@syr.edu

**Data Availability Statement:** All relevant data are within the paper and its Supporting Information files. The data underlying the results presented in

## Abstract

The gut microbiome may affect overall cardiometabolic health. Enterolactone is an entero-lignan reflective of dietary lignan intake and gut microbiota composition and diversity that can be measured in the urine. The purpose of this study was to examine the association between urinary enterolactone concentration as a reflection of gut health and blood pressure/risk of hypertension in a large representative sample from the US population. This analysis was conducted using data from the National Health and Nutrition Examination Survey (NHANES) collected from January 1999 through December 2010. Variables of interest included participant characteristics (including demographic, anthropometric and social/environmental factors), resting blood pressure and hypertension history, and urinary enterolactone concentration. 10,637 participants (45 years (SE = 0.3), 51.7% (SE = 0.6%) were female) were included in analyses. In multivariable models adjusted for demographic, socioeconomic and behavioral/environmental covariates, each one-unit change in log-transformed increase in enterolactone was associated with a 0.738 point (95% CI: -0.946, -0.529; p<0.001) decrease in systolic blood pressure and a 0.407 point (95% CI: -0.575, -0.239; p<0.001) decrease in diastolic blood pressure. Moreover, in fully adjusted models, each one-unit change in log-transformed enterolactone was associated with 8.2% lower odds of hypertension (OR = 0.918; 95% CI: 0.892, 0.944; p<0.001). Urinary enterolactone, an indicator of gut microbiome health, is inversely associated with blood pressure and hypertension risk in a nationally representative sample of U.S. adults.

## Introduction

Hypertension, or high blood pressure, is a global epidemic [1], affecting 32% of women and 34% of men worldwide [2]. Hypertension is associated with damage to vital target organs, contributing to higher rates of heart failure, kidney failure, nonalcoholic fatty liver disease, cognitive decline, myocardial infarction, and stroke [3]. Blood pressure is modifiable and can be improved by lifestyle factors such as diet. Increasing the consumption of lignans may be

the study are also available from the National Health and Nutrition Examination Survey.

**Funding:** The author(s) received no specific funding for this work.

**Competing interests:** The authors have declared that no competing interests exist.

particularly advantageous for reducing hypertension risk. Lignans are a type of phytoestrogen and are bioactive, phenolic plant compounds found in highest concentration in flax and sesame seeds, with lower concentrations found in other seeds, grains, fruits, and vegetables. Once the lignan is digested, bacteria in the gut metabolizes the compound into postbiotic enterolignans, which includes enterolactone and enterodiol [4–6]. Concentrations of these enterolignans can be measured in the blood or urine [4,5]. Thus, enterolignans are emerging as important inferential biomarkers of health that reflect not only dietary lignan intake, but also gut microbiota composition and diversity [7].

Enterolactone is the main circulating enterolignan [4]. Enterolactone levels measured in blood and/or urine are inversely associated with inflammation, cardiometabolic and coronary disease, and cardiovascular and all-cause mortality [8–12]. Blood pressure is an important CVD risk factor, however its association with enterolactone is incompletely understood. Small-scale studies note inverse associations between serum enterolactone and hypertension risk [1]. In order to further substantiate the association of enterolactone as a potential biomarker of gut health and CVD risk, we set out to examine the association between urinary enterolactone concentration and blood pressure/risk of hypertension in a large representative sample from the US population. We further accounted for participant demographic characteristics, biobehavioral factors, and environmental factors that may influence this association. We hypothesized that increased urinary enterolactone concentration would be inversely associated with both blood pressure and the risk of hypertension.

## Materials and methods

### Design

This study involved use of serial cross-sectional data from the first six continuous waves of the National Health and Nutrition Examination Survey (NHANES) collected from January 1999 through December 2010. The NHANES is based on a nationally representative sample of the non-institutionalized population of the United States. The data contains measures concurrently obtained from questionnaires, physical examinations, and urinary biomarker specimens. NHANES procedures have been ethically approved by the National Center for Health Statistics Ethics Review Board. Prior to any data collection, signed participant consent was obtained from all individuals. NHANES participant data are de-identified and are publicly available. Additional information on NHANES methodology and data collection can be obtained from the NHANES website (https://www.cdc.gov/nchs/nhanes.htm). Further, all NHANES survey response rates have been published elsewhere (https://wwwn.cdc.gov/nchs/nhanes/ResponseRates.aspx). The study was reviewed by the University of Texas at San Antonio Institutional Review Board and was determined to be research not involving human subjects as defined in 45 CFR 46.104(3)(A).

### Participants

Adult respondents 18 years of age and older were included in this study. Phytoestrogens, including enterolactone, were obtained from urine samples on 1/3 random sample of NHANES respondents. As a result, this study was limited to the phytoestrogen sub-sample that included measurements for both enterodiol and enterolactone (n = 10,637), and 2/3 of the total sample (n = 24,693) was excluded. In analyses of continuous blood pressure measures, 420 cases were excluded due to lack of valid blood pressure data.

## Measures

The dependent variables for this study were continuous systolic and diastolic blood pressure readings and categorical indication of hypertension (yes/no). Three consecutive auscultatory blood pressure readings were taken for each participant after a 5-minute resting period [13], from which we calculated the average of 3 systolic and diastolic blood pressure readings. Ascertainment of hypertension involved a composite approach based on any indication of hypertension from (1) blood pressure readings based on the current American Heart Association guidelines (Hypertension defined as systolic blood pressure $\geq$130 mmHg or diastolic blood pressure $\geq$ 80 mmHg) [14], (2) self-reported diagnosis of hypertension by a healthcare professional, or (3) self-reported use of blood pressure medication. Blood pressure readings are reported as mmHg.

Urinary enterolactone levels were assessed by enzymatic deconjugation with solid-phase extraction and reverse-phase high-performance liquid chromatography [15,16]. Enterolactone measurements are reported in ng/ml and were transformed using the natural log for analytical purposes.

Covariates for this study included demographic, socioeconomic, anthropometric, and behavioral measures. Demographic variables included age (as continuous variable), self-identified sex (male or female [reference]), race/ethnicity (Mexican American, Other Hispanic, non-Hispanic Black, non-Hispanic White [reference], and Other Race (including multi-racial)), and marital status (married/living with partner, widowed/divorced/separated, never married [reference], or missing). Socioeconomic variables included household income level (<$20,000 per year[reference], $20,000 to $54,999, $55,000 or more, >$20,000 but unspecified amount, or missing) and educational attainment (less than high school [reference], high school graduate or equivalent, some college, college graduate and more, or missing). Behavior and anthropometric variables included smoking status (former smoker, current smoker, never smoked [reference]), participation in moderate or vigorous activities in the past 30 days (yes, no [reference], missing), body mass index (18.5 and lower (underweight), 18.6 to 24.9 (normal) [reference], 25 to 29.9 (overweight), 30 and above (obese), and missing). Self-reported use of antibiotic medications (yes or none reported [reference]) was also included as a possible confounder due to prior findings showing an association between antibiotics and enterolactone levels [17].

## Statistical analysis

Descriptive statistics are reported as mean and standard error (SE) for continuous variables, or as percent and SE for categorical variables, and were compared using t-tests for continuous data and Rao-Scott adjusted chi-square for categorical data. Enterolactone did not follow a normal distribution and was transformed for analysis by taking the natural log of the raw enterolactone measure, which approximated a normal distribution. Continuous systolic and diastolic blood pressure data were analyzed using unadjusted and multivariable adjusted linear regression models, and are reported as coefficient, 95% confidence interval (CI) and p-values. Binary hypertension data were analyzed using unadjusted and multivariable adjusted logistic regression models, and are reported as odds ratios (OR), 95% CI, and p-values. All data were analyzed with adjustments for complex sample design and weighting. Estimates of mean systolic and diastolic blood pressure and hypertension prevalence from multivariable adjusted models were plotted with 95% prediction intervals (PI). Data were analyzed using IBM SPSS Statistics version 27 (Chicago, IL), and data visualizations were performed using R version 4.1.1 (R Foundation for Statistical Computing).

## Results

A total of 10,637 participants were included, with a mean age of 45 years (SE = 0.3), 51.7% (SE = 0.6%) were female, and 70.2% (SE = 1.3%) were of non-Hispanic White racial/ethnic background (Table 1). The weighted total prevalence of hypertension was 46.3% (SE = 0.7%). Of the individuals with hypertension (either systolic > = 130 mmHg or diastolic > = 80 mmHg or self-reported diagnosis by a healthcare provider or self-reported taking blood

**Table 1. Weighted descriptive statistics by composite hypertension indicator (n = 10,637).**

| Variables | Total Sample n = 10,637 | Hypertension n = 5,172 | No Hypertension n = 5,465 | p-value |
|---|---|---|---|---|
| Enterolactone (natural log transformed) ng/mL, mean (SE) | 5.5 (0.03) | 5.4 (0.03) | 5.6 (0.04) | <0.001 |
| Age, mean (SE) | 45.0 (0.3) | 53.3 (0.4) | 37.9 (0.3) | <0.001 |
| Age Groups, percent (SE) | | | | |
| 18–34 | 31.8 (0.7) | 14.2 (0.8) | 46.9 (1.0) | <0.001 |
| 35–49 | 30.2 (0.7) | 27.8 (0.9) | 32.2 (0.9) | |
| 50–64 | 22.5 (0.6) | 30.7 (0.8) | 15.4 (0.7) | |
| 65 and older | 15.6 (0.5) | 27.3 (0.9) | 5.5 (0.3) | |
| Sex, percent (SE) | | | | |
| Female | 51.7 (0.6) | 47.2 (0.9) | 55.6 (0.8) | <0.001 |
| Male | 48.3 (0.6) | 52.8 (0.9) | 44.4 (0.8) | |
| Race/Ethnicity, percent (SE) | | | | |
| Mexican American | 7.9 (0.6) | 5.6 (0.6) | 9.9 (0.7) | <0.001 |
| Other Hispanic | 5.2 (0.8) | 4.1 (0.7) | 6.1 (0.8) | |
| Non-Hispanic Black | 11.3 (0.7) | 13.0 (0.9) | 9.8 (0.7) | |
| Non-Hispanic White | 70.2 (1.3) | 72.4 (1.4) | 68.4 (1.4) | |
| Other | 5.3 (0.4) | 4.9 (0.5) | 5.7 (0.5) | |
| Marital Status, percent (SE) | | | | |
| Married/Living with Partner | 61.3 (0.8) | 63.5 (0.9) | 59.3 (1.1) | <0.001 |
| Widowed/Divorced/Separated | 17.6 (0.6) | 23.0 (0.9) | 12.9 (0.6) | |
| Never Married | 18.2 (0.7) | 11.2 (0.6) | 24.2 (1.0) | |
| Missing | 3.0 (0.6) | 2.2 (0.7) | 3.6 (0.5) | |
| Education, percent (SE) | | | | |
| Less than High School Graduate | 19.8 (0.7) | 21.0 (0.9) | 18.7 (0.7) | 0.005 |
| High School Graduate/or Equivalent | 25.6 (0.7) | 27.0 (1.0) | 24.4 (1.0) | |
| Some College/Associates Degree | 30.6 (0.7) | 29.8 (0.9) | 31.4 (0.9) | |
| College Graduate or More | 23.9 (0.9) | 22.1 (1.0) | 25.4 (1.0) | |
| Missing | 0.1 | | | |
| Income to Poverty Ratio, percent (SE) | | | | |
| At or Below Poverty | 13.4 (0.7) | 12.4 (0.8) | 14.3 (0.7) | 0.01 |
| 1 to 3 Times Above Poverty | 33.3 (0.8) | 35.0 (1.0) | 31.8 (1.0) | |
| >3 Times Above Poverty | 46.3 (1.0) | 45.6 (1.3) | 46.9 (1.1) | |
| Missing | 7.0 (0.4) | 7.0 (0.5) | 7.0 (0.5) | |
| Smoking Status, percent (SE) | | | | |
| Current Smoker | 23.0 (0.7) | 20.1 (0.9) | 25.5 (0.9) | <0.001 |
| Former Smoker | 23.3 (0.6) | 28.4 (0.8) | 18.9 (0.9) | |
| Never Smoked | 50.0 (0.8) | 50.4 (0.9) | 49.6 (1.2) | |
| Missing | 3.7 (0.2) | 1.1 (0.2) | 5.9 (0.3) | |
| Body Mass Index, percent (SE) | | | | |
| 18.5 and lower (Underweight) | 1.9 (0.2) | 1.2 (0.2) | 2.5 (0.3) | <0.001 |
| 18.6 to 24.9 (Normal Weight) | 31.6 (0.7) | 22.4 (0.8) | 39.5 (0.8) | |
| 25 to 29.9 (Overweight) | 31.3 (0.6) | 31.4 (0.7) | 31.2 (0.9) | |
| 30 and above (Obese) | 32.8 (0.7) | 42.6 (0.9) | 22.4 (0.8) | |
| Missing | 2.4 (0.2) | 2.4 (0.3) | 2.5 (0.2) | |
| Moderate Physical Activity Past 30 Days, percent (SE) | | | | |
| Yes | 35.3 (0.9) | 33.4 (1.0) | 36.9 (1.1) | <0.001 |
| No | 29.5 (0.9) | 31.1 (1.1) | 28.1 (1.0) | |
| Unable | 1.4 (0.1) | 2.3 (0.3) | 0.7 (0.1) | |
| Missing | 33.8 (1.3) | 33.2 (1.6) | 34.3 (1.4) | |
| Antibiotic Use, percent (SE) | | | | |
| Yes | 1.4 (0.2) | 1.5 (0.2) | 1.3 (0.2) | 0.53 |
| None Reported | 98.6 (0.2) | 98.5 (0.2) | 98.7 (0.2) | |

pressure medication), 76.1% (SE = 0.8%) had measured hypertension at time of examination (systolic > = 130 mmHg or diastolic > = 80 mmHg), 58.4% (SE = 1.0%) reported being told by a healthcare provider that they had high blood pressure, and 46.7% (SE = 1.0%) reported taking blood pressure medication. Individuals with hypertension had a higher mean age (53.3 years vs. 37.9 years; p<0.001), were more likely to be male (52.8% vs. 44.4%; p<0.001) and more likely to be non-Hispanic Black (13.0% vs. 9.8%; p<0.001) or non-Hispanic White (72.4% vs. 68.4%; p<0.001). Individuals with hypertension were also more likely to be married/living with partner (63.5% vs. 59.3%; p<0.001) or widowed/divorced/separated (23.0% vs. 12.9%; p<0.001). Individuals with hypertension were less likely to have a college education (22.1% vs. 25.4%; p = 0.005), live at or below poverty (12.4% vs. 14.3%; p = 0.01), be a current smoker (20.1% vs. 25.5%; p<0.001), have a body mass index in the 'normal range' of 18.6 to 24.9 (22.4% vs. 39.5%; p<0.001) and to have engaged in moderate physical activity in the past 30 days (33.4% vs. 36.9%; p<0.001). Mean enterolactone (natural log transformed) was also lower for individuals with hypertension (5.4 vs. 5.6; p<0.001).

In linear regression models of continuous systolic and diastolic blood pressure, log-transformed enterolactone was inversely associated with blood pressure, in unadjusted and multivariable adjusted models (**Table 2**). In unadjusted models, each one-unit change in log-transformed increase in enterolactone was associated with a 0.582 point (95% CI: -0.817, -0.347; p<0.001) lower systolic and a 0.366 point (95% CI: -0.520, -0.211; p<0.001) lower diastolic blood pressure. In multivariable models adjusted for demographic, socioeconomic and behavioral covariates, the association between enterolactone and blood pressure was not attenuated, rather it strengthened. Each one-unit change in log-transformed increase in enterolactone was associated with a 0.738 point (95% CI: -0.946, -0.529; p<0.001) decrease in systolic blood pressure and a 0.407 point (95% CI: -0.575, -0.239; p<0.001) decrease in diastolic blood pressure. **Fig 1** illustrates the association between enterolactone and mean blood pressure levels, showing a steady decrease in systolic and diastolic blood pressure as enterolactone increases. For example, at 0.5 log ng/ml (1.65 ng/ml unlogged) mean systolic blood pressure is 125.7 (95% PI: 123.7, 127.8) and at 7 log ng/ml (1096.6 ng/ml unlogged) mean systolic blood pressure falls to 120.9 (95% PI: 117.5, 124.4). Similarly, mean diastolic blood pressure is 72.6 (95% PI: 70.5, 74.7) at 0.5 log ng/ml, and falls to 69.9 (95% PI: 66.7, 73.1) at 7 log ng/ml (1096.6 ng/ml unlogged). Age, sex, race, ethnicity, marital status, education, smoking status, and BMI were also associated with systolic and diastolic blood pressure levels (**Table 2**).

In unadjusted and multivariable adjusted logistic regression models of hypertension, log-transformed enterolactone was associated with lower odds of hypertension (**Table 3**). In the unadjusted model, each one-unit change in log-transformed enterolactone was associated with 5.4% lower odds of hypertension (OR = 0.946; 95% CI: 0.923, 0.970; p<0.001). Like the models of continuous blood pressure, the enterolactone-hypertension association was strengthened in the multivariable adjusted model. Each one-unit change in log-transformed enterolactone was associated with 8.2% lower odds of hypertension (OR = 0.918; 95% CI: 0.892, 0.944; p<0.001). The probability of hypertension decreases steadily as enterolactone levels increase (**Fig 2**). For example, at 0.5 log ng/ml (1.65 ng/ml unlogged) the probability of hypertension was 0.57 (95% PI: 0.47, 0.66) and at 7 log ng/ml (1096.6 ng/ml unlogged) the probability of hypertension is 0.43 (95% PI: 0.30, 0.57). Age, sex, race, ethnicity, smoking status, BMI and self-reported antibiotic use were also associated with hypertension (**Table 3**).

## Discussion

The novel findings of this study are twofold: 1) Higher enterolactone concentration was associated with lower odds of hypertension and lower systolic and diastolic blood pressure in a large

**Table 2. Results of weighted, multivariable regression analyses of enterolactone by blood pressure (n = 10,217).**

| Variables | Systolic Blood Pressure Coefficient (95% CI); p value | Diastolic Blood Pressure Coefficient (95% CI); p-value |
|---|---|---|
| **Unadjusted:** | | |
| Enterolactone (natural log transformed) | -0.582 (-0.817, -0.347); <0.001 | -0.366 (-0.520, -0.211); <0.001 |
| **Adjusted:** | | |
| Enterolactone (natural log transformed) | -0.738 (-0.946, -0.529); <0.001 | -0.407 (-0.575, -0.239); <0.001 |
| Age | 0.540 (0.514, 0.567); <0.001 | -0.015 (-0.038, 0.008); 0.198 |
| Sex | | |
| Female (ref) | | |
| Male | 4.311 (3.611, 5.010); <0.001 | 3.613 (2.985, 4.242); <0.001 |
| Race/Ethnicity | | |
| Mexican American | -0.291 (-1.185, 0.603); 0.520 | -1.555 (-2.544, -0.566); 0.002 |
| Other Hispanic | -0.896 (-2.457, 0.664); 0.257 | -0.042 (-1.261, 1.177); 0.945 |
| Non-Hispanic Black | 3.481 (2.562, 4.401); <0.001 | 1.539 (0.504, 2.574); 0.004 |
| Non-Hispanic White (ref) | | |
| Other | 1.760 (-0.097, 3.618); 0.063 | 1.492 (-0.077, 3.061); 0.062 |
| Marital Status | | |
| Married/Living with Partner | -2.295 (-3.046, -1.544); <0.001 | 3.114 (2.231, 3.997); <0.001 |
| Widowed/Divorced/Separated | -0.990 (-2.305, 0.325); 0.138 | 2.893 (1.861, 3.925); <0.001 |
| Never Married (ref) | | |
| Missing | -1.720 (-4.791, 1.351); 0.269 | 1.265 (-1.376, 3.906); 0.344 |
| Education | | |
| Less than High School Graduate (ref) | | |
| High School Graduate/or Equivalent | 0.289 (-0.900, 1.477); 0.631 | 1.384 (0.474, 2.295); 0.003 |
| Some College/Associates Degree | -0.474 (-1.771, 0.824); 0.470 | 1.674 (0.772, 2.576); <0.001 |
| College Graduate or More | -2.630 (-3.991, -1.268); <0.001 | 1.703 (0.633, 2.773); 0.002 |
| Missing | -0.219 (-17.132, 16.694); 0.980 | 7.434 (-3.131, 17.998); 0.166 |
| Income to Poverty Ratio, percent (SE) | | |
| At or Below Poverty (ref) | | |
| 1 to 3 Times Above Poverty | -0.741 (-2.270, 0.787); 0.338 | -1.335 (-2.420, -0.250); 0.016 |
| >3 Times Above Poverty | -1.352 (-2.770, 0.067); 0.062 | 0.170 (-0.989, 1.328); 0.772 |
| Missing | -0.406 (-2.237, 1.424); 0.660 | |
| Smoking Status | | |
| Current Smoker | -1.599 (-2.521, -0.677); <0.001 | -0.910 (-2.379, 0.558); 0.221 |
| Former Smoker | -1.830 (-2.828, -0.832); <0.001 | -0.995 (-1.813, -0.177); 0.018 |
| Never Smoked (ref) | | -0.560 (-1.410, 0.290); 0.194 |
| Missing | | |
| Body Mass Index | 2.730 (1.242, 4.218); <0.001 | -5.080 (-6.783, -3.376); <0.001 |
| 18.5 and lower (Underweight) | | |
| 18.6 to 24.9 (Normal Weight) (ref) | -1.520 (-5.420, 2.381); 0.441 | 1.802 (-0.579, 4.183); 0.136 |
| 25 to 29.9 (Overweight) | | |
| 30 and above (Obese) | 1.884 (0.954, 2.815); <0.001 | 1.249 (0.517, 1.981); 0.001 |
| Missing | 4.077 (3.147, 5.007); <0.001 | 3.062 (2.290, 3.835); <0.001 |
| Moderate Physical Activity Past 30 Days | 1.510 (-1.940, 4.960); 0.387 | 0.049 (-2.258, 2.356); 0.967 |
| Yes | | |
| No (ref) | 0.222 (-0.618, 1.062); 0.601 | -0.441 (-1.349, 0.467); 0.337 |
| Unable | | |
| Missing | -0.429 (-4.396, 3.537); 0.830 | -0.987 (-3.935, 1.961); 0.508 |
| Antibiotic Use | -2.330 (-3.278, -1.381); <0.001 | -1.730 (-2.911, -0.550); 0.005 |
| Yes | | |
| None Reported (ref) | 3.081 (-0.217, 6.380); 0.067 | 0.271 (-2.219, 2.760); 0.83 |

representative cohort of U.S. adults, and 2) accounting for age, sex, race, ethnicity, marital status, education, smoking status, and BMI strengthened, rather than attenuated, the association between enterolactone and hypertension, as well as the relationship between enterolactone and blood pressure. Therefore, enterolactone concentration may be a novel predictor of elevated blood pressure and hypertension risk, which is strengthened when accounting for participant demographic characteristics and environmental factors.

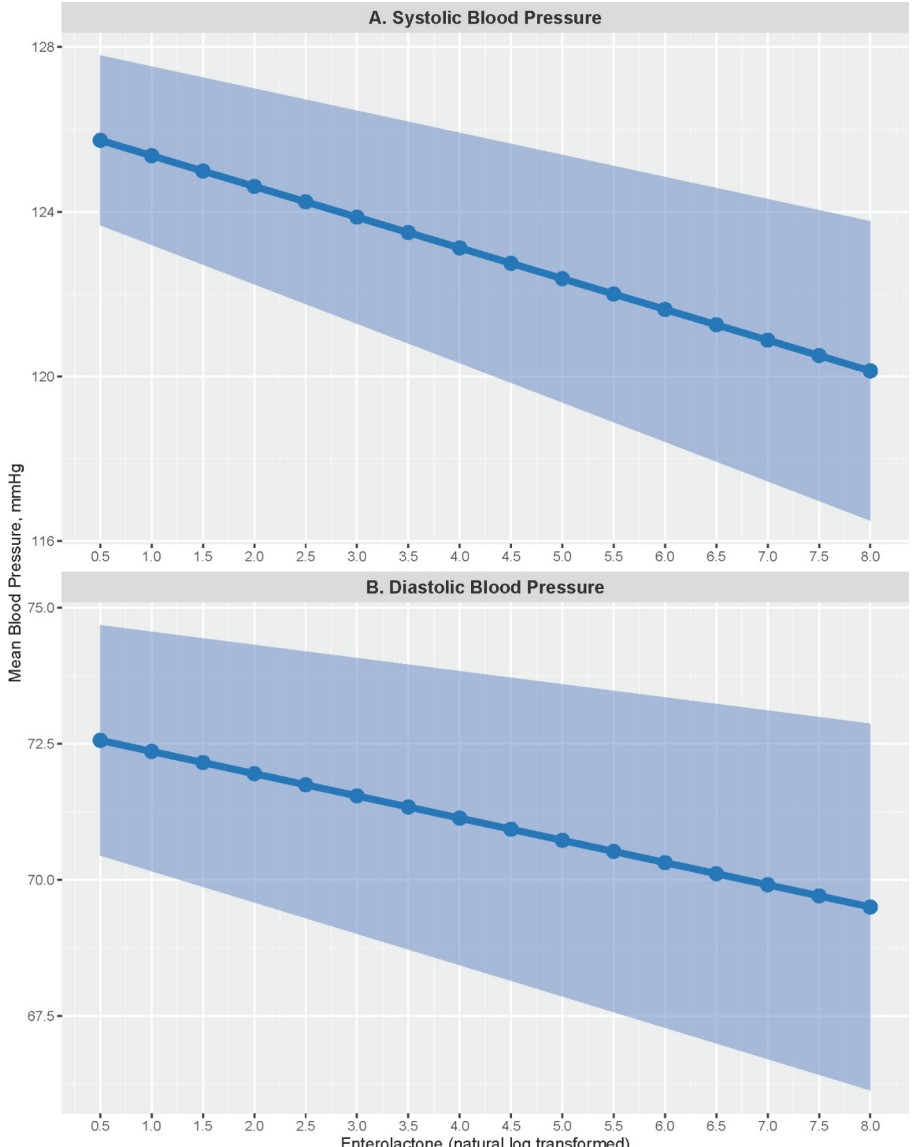

**Fig 1.** Multivariable adjusted estimates of mean systolic (A) and diastolic (B) blood pressure levels at varying levels of log-transformed enterolactone. Shaded area represents the 95% prediction interval.

Higher enterolactone concentration may be a predictor of lower blood pressure due to its association with the gut microbiota. Gut microorganisms are responsible for the metabolism of lignans to enterolignans, and high urinary concentrations of enterolactone suggest favorable gut microbial environment. High enterolactone levels have been shown to be inversely associated with cardiometabolic disease [8,9] and cardiovascular disease [11] and have been suggested to mediate the relationship between diet quality and cardiometabolic health [18]. Our findings support those of Lee et al. [1], in which higher tertiles of enterolactone concentration were associated with lower prehypertension and hypertension risk among 229 pre/hypertensive and 159 healthy Korean adults [1]. In the current study which used a sample of over 10,000 U.S. adults from NHANES, urinary enterolactone concentration was significantly lower in hypertensive individuals compared to non-hypertensive individuals and lower levels

**Table 3. Results of weighted, unadjusted and multivariable adjusted logistic regression analyses of composite hypertension (n = 10,637).**

| Variables | OR (95% CI); p-value |
|---|---|
| **Unadjusted:** | |
| Enterolactone (natural log transformed) | 0.946 (0.923, 0.970); <0.001 |
| **Adjusted:** | |
| Enterolactone (natural log transformed) | 0.918 (0.892, 0,944); <0.001 |
| Age | 1.068 (1.064, 1.072); <0.001 |
| Sex | |
| Female (ref) | |
| Male | 1.805 (1.610, 2.024); <0.001 |
| Race/Ethnicity | |
| Mexican American | 0.729 (0.623, 0.854); <0.001 |
| Other Hispanic | 0.712 (0.577, 0.879); 0.002 |
| Non-Hispanic Black | 1.537 (1.295, 1.823); <0.001 |
| Non-Hispanic White (ref) | |
| Other | 1.189 (0.890, 1.588); 0.239 |
| Marital Status | |
| Married/Living with Partner | 0.971 (0.825, 1.141); 0.715 |
| Widowed/Divorced/Separated | 0.993 (0.803, 1.229); 0.948 |
| Never Married (ref) | |
| Missing | 1.035 (0.680, 1.576); 0.870 |
| Education | |
| Less than High School Graduate (ref) | |
| High School Graduate/or Equivalent | 1.129 (0.958, 1.330); 0.146 |
| Some College/Associates Degree | 1.114 (0.930, 1.334); 0.237 |
| College Graduate or More | 0.947 (0.772, 1.160); 0.593 |
| Missing | 0.627 (0.045, 8.768); 0.726 |
| Income to Poverty Ratio, percent (SE) | |
| At or Below Poverty (ref) | |
| 1 to 3 Times Above Poverty | 0.810 (0.680, 0.965); 0.019 |
| >3 Times Above Poverty | 0.730 (0.612, 0.871); <0.001 |
| Missing | 0.649 (0.514, 0.821); <0.001 |
| Smoking Status | |
| Current Smoker | 0.827 (0.711, 0.962); 0.014 |
| Former Smoker | 0.859 (0.741, 0.995); 0.043 |
| Never Smoked (ref) | |
| Missing | 0.909 (0.633, 1.305); 0.601 |
| Body Mass Index | 0.949 (0.561, 1.607); 0.845 |
| 18.5 and lower (Underweight) | |
| 18.6 to 24.9 (Normal Weight) (ref) | 1.385 (1.195, 1.604); <0.001 |
| 25 to 29.9 (Overweight) | 2.797 (2.418, 3.234); <0.001 |
| 30 and above (Obese) | 1.197 (0.834, 1.716); 0.325 |
| Missing | |
| Moderate Physical Activity Past 30 Days | 0.948 (0.819, 1.097); 0.465 |
| Yes | |
| No (ref) | 1.310 (0.789, 2.177); 0.293 |
| Unable | 0.805 (0.700, 0.926); 0.003 |
| Missing | |
| Antibiotic Use | 1.399 (1.039, 1.882); 0.027 |
| Yes | |
| None Reported (ref) | |

were predictive of higher systolic blood pressure, diastolic blood pressure and hypertension status. The results of the current analysis conflict with a previous study by Frankenfeld et al. [8]. In this previous study that also utilized NHANES, higher enterolactone concentration was not associated with high blood pressure (defined as a categorical variable). Discrepancy may be related to differences in sample size (n = 2,260 adults in Frankenfeld et al.) as well as differences in independent variables that were included and/or excluded in statistical models. Previous work did not consider physical activity status, body mass index, marital status, and

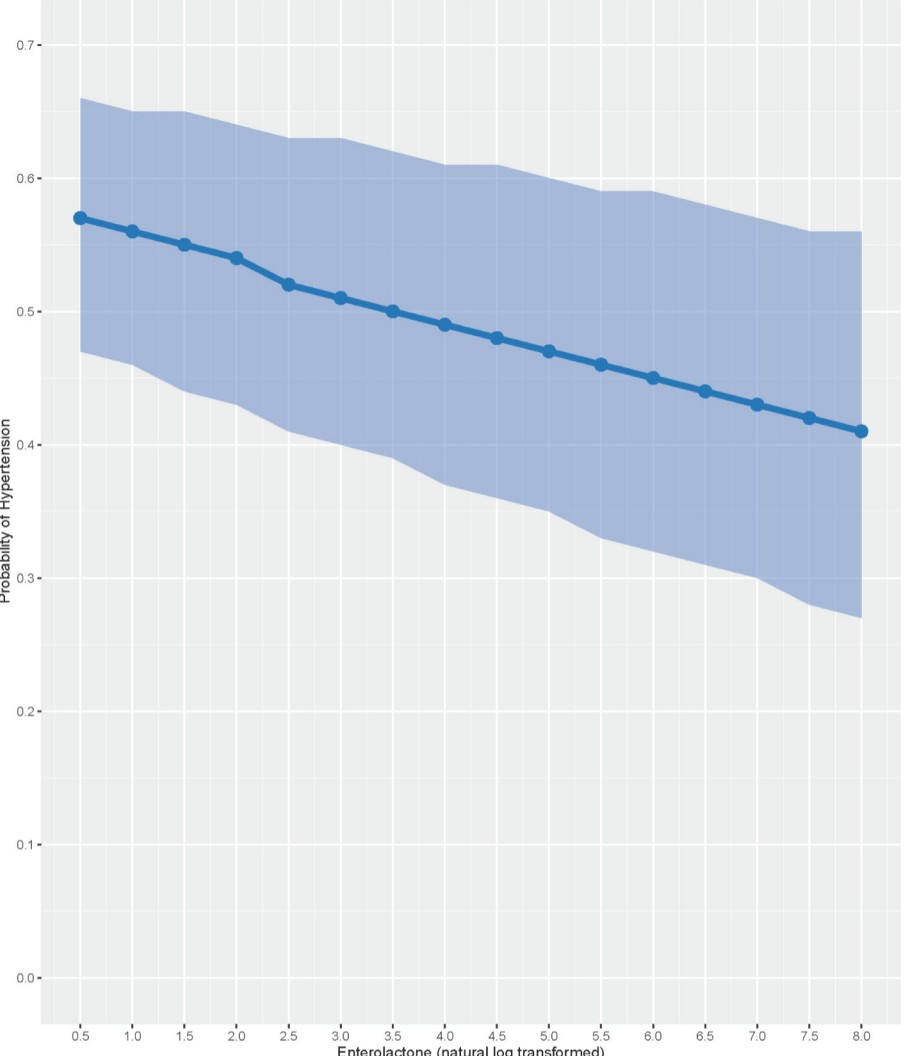

**Fig 2. Multivariable adjusted probability of hypertension at varying levels of log-transformed enterolactone.** Shaded area represents the 95% prediction interval.

income-to-poverty ratio, all of which may impact not only blood pressure, but diet and gut health. Indeed, we noted that the addition of these demographic and biobehavioral factors to our models strengthened, rather than attenuated, associations between enterolactone levels and blood pressure. Taken together and our findings support an inverse association between enterolactone and blood pressure / hypertension in a large sample of U.S. adults.

Inclusion of various demographic, biobehavioral and lifestyle factors strengthened, rather than attenuated, the association between enterolactone and blood pressure. This may be taken to suggest that these factors are moderators of the association between enterolactone and blood pressure as they may impact overall hypertension risk as well as gut microbiota composition. For example, male sex, Black race, higher BMI, older age, and smoking increase the risk for hypertension [19–24]. These same demographic, environmental and lifestyle factors also impact the gut microbiota. Cigarettes are composed of multiple chemicals, each with their own detrimental effect on the microbiome [25]. Higher BMI and obesity are also associated with changes in the composition of the gut microbiota, which are known to contribute the

development of obesity [26,27]. Since smoking and obesity result in unfavorable changes in gut microbiota, they are all also associated with a decrease in enterolactone concentration [28–31]. In contrast, physical activity, higher education, higher income, and being married is associated with improved vascular outcomes and decreased risk for hypertension [3,32–35]. Physical activity and habitual exercise may have a favorable effect on the gut microbiome [36,37]. A study by Xia et al. found spontaneous hypertensive rats had a significant decrease in systolic blood pressure following a 12-week moderate exercise intervention that correlated with a decreased abundance of several pathogenic microbial strains [38,39]. Higher education and income may result in higher health literacy coupled with an enriched living environment enabling access to healthier foods, promoting overall gut health. Thus, modifiable CVD risk factors may also be modifiable gut health factors.

In addition to modifiable risk factors, there are non-modifiable risk factors that may influence the gut microbiota and thus risk for hypertension. For example, aging is associated with altered gut microbiota composition with the loss of key taxa over time [40] as well as the addition of variants related to disease [40–42]. There are notable sex differences in gut microbiome composition in both humans [43,44] and animals [45,46] and gut microbiome dysregulation may be more strongly associated with blood pressure and hypertension risk in women compared to men [47]. There may be racial variation in gut microbiome composition between White and Black hypertensive individuals [48]. However, it should be noted that some racial variation may be driven by systemic environmental factors related to socioeconomic status like education and wealth that can influence lifestyle and thus gut microbiome composition [49,50]. Taken together, our findings highlight that numerous interactive individual-level factors may moderate associations between enterolactone and blood pressure.

Our findings are biologically plausible due to the possible effects of enterolactone on blood pressure regulation. Lignan-derived polyphenols such as enterolactone function as antioxidants and are known to attenuate excess reactive oxygen species (ROS) production, thus reducing oxidative stress [51]. This is important for maintaining the bioavailability of nitric oxide (NO), a potent vasodilator that is needed to promote vascular endothelial function and maintain optimal blood pressure [52]. In addition, enterolactone is a phytoestrogen, with has been shown to have weak estrogenic properties [53], including activating tissue-specific estrogen receptors [54]. Estrogen is known to activate endothelial nitric oxide synthase via nongenomic estrogen receptor alpha, thus increasing the production of NO [55]. However, whether enterolactone acts similarly to endogenous estrogen, stimulating NO production via endothelial nitric oxide synthase, has yet to be determined.

High enterolactone concentration is correlated with greater gut microbial diversity, which may influence the development of hypertension [38]. Increased gut microbial diversity is associated with enhanced production of short chain fatty acids (SCFA) such as acetate, butyrate, and propionate, which play integral roles in maintaining host biological processes, including blood pressure regulation [56]. Supplementation with SCFA may be a promising treatment of endothelial dysfunction. For example, supplementation with acetate has been shown to improve endothelial function and aortic stiffness in mice [57,58]. This is important to note as age-associated increase in aortic stiffness and loss of endothelial-dependent vasodilation as manifestations of vascular dysfunction are established precursors to the development of hypertension [59,60]. Parenthetically, age-related gut dysbiosis significantly contributes to vascular dysfunction [61] and age-related aortic stiffness and endothelial dysfunction can be ameliorated via fecal microbiota transplant (FMT) from young healthy mice to older mice [58,62]. Moreover, a study by Li et al. showed high blood pressure to be transferable between hypertensive humans and germ-free mice via fecal transplantation, further illuminating a unique causal role for gut microbial diversity as a risk factor for hypertension [14]. Taken together and these

studies support that age-related changes to the gut microbiome with subsequent decreases in SCFA and other metabolites like enterolignans may negatively affect vascular function and affect blood pressure regulation.

The composition of the gut microbiome impacts the ability to process lignans. Intervention studies have indicated that the foundation of the microbiome drives an individual's capability of converting lignan-rich dietary sources to enterolactone as observed in both serum and urine levels [6,63–65]. Lampe et al. [6] demonstrated high enterolactone producing healthy adults aged 20–45 years have a more robust gut microbial diversity compared to low enterolactone excreters, independent of 60-day lignan supplementation. Additionally, Lagkouvardos et al. [64] observed increased serum enterolactone following a one-week dietary intervention but no change to dominant bacterial makeup in nine healthy males. A study conducted by Hullar et al. [66] found a positive correlation between urinary enterolactone excretion and gut microbial diversity. It appears that while additional lignan sources to the diet do increase serum and urinary enterolactone levels, the ability to produce enterolactone is either impeded or accelerated based on individual microbial diversity. For example, members of the genus *Klebsiella* are more abundant in both non-enterolactone producers [7] and individuals with hypertension [67]. Conversely, both Sawane et al. and Lampe et al. observed a more robust abundance of the *Ruminococcaceae* genera (Firmicutes phylum) in enterolactone producers, regardless of dietary intervention [6,7]. Thus, greater urinary enterolactone levels are not only a reflection of dietary lignan intake, but are a reflection of host gut microbial diversity.

The current study includes some limitations. We did not account for individual differences in diet nor directly assess microbiome composition in the current analysis. In addition, recent literature suggest menopause may impact decrease a women's gut microbial diversity due to the reduction of sex hormones, whereas prior to menopause, women have increased diversity compared to men [68]. Therefore, this results in sex differences in the microbiome with aging [68]. In addition, menopause is also associated with an increase in blood pressure and cardiovascular disease risk [69]. Therefore, whether menopause status among women included in this analysis affects the association between urinary enterolactone concentration and blood pressure/hypertension risk is unknown, particularly when age is included in the model. Future studies should account for menopause status to determine if this may confound these associations. This was a cross sectional study and as such, causation cannot be inferred. Much of our discussion considered the effect of enterolactone on blood pressure but it is also possible that hypertension may alter gut health [70]. High blood pressure may cause intestinal damage and increase gut permeability [71]. Moreover, prescribed antihypertensive medications share a bidirectional relationship with the gut microbiome [72]. The composition of the gut microbiome influences the pharmacokinetics of antihypertensive medications and these medications alter microbial diversity [72]. This bi-directional relationship between the gut microbiome and antihypertensive medications also differs across the type of medication prescribed, such as a calcium-channel blocker versus an angiotensin II receptor blocker [72]. Additionally, antibiotic use is associated with lower enterolactone levels 3–12 months following course completion, further highlighting the impact of microbe diversity on lignin metabolism [17,73]. This compounded with the basal capacity to produce enterolactone could elevate one's risk of developing or exacerbating hypertension. Taken together, our findings should be interpreted with care–enterolactone levels may be a reflection of gut dysbiosis as either a cause or consequence of hypertension and its related sequela like obesity, poor diet, and physical inactivity.

In conclusion, urinary enterolactone may be a novel indicator for gut microbiome health and hypertension risk in the general U.S population. This indicator is also strengthened, rather than attenuated, by the inclusion of demographic, social and environmental factors,

highlighting their moderating effect on the gut microbiome as well as blood pressure and hypertension risk.

## Supporting information

**S1 Checklist.**
(DOCX)

**S1 File.**
(XLSX)

## Author Contributions

**Conceptualization:** Jeffrey T. Howard, Kevin S. Heffernan.

**Formal analysis:** Jeffrey T. Howard.

**Supervision:** Jeffrey T. Howard, Kevin S. Heffernan.

**Writing – original draft:** Cynthia M. Weiner, Shannon E. Khan, Caleb Leong, Sushant M. Ranadive, Sara C. Campbell, Jeffrey T. Howard, Kevin S. Heffernan.

**Writing – review & editing:** Cynthia M. Weiner, Shannon E. Khan, Sushant M. Ranadive, Sara C. Campbell, Jeffrey T. Howard, Kevin S. Heffernan.

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
