## [Decision Letter · Decision Letter 0]

4 Dec 2023

PONE-D-23-26257Association of enterolactone with blood pressure and hypertension risk in NHANESPLOS ONE

Dear Dr. Heffernan,

Thank you for submitting your manuscript to PLOS ONE. After careful consideration, we feel that it has merit but does not fully meet PLOS ONE’s publication criteria as it currently stands. Therefore, we invite you to submit a revised version of the manuscript that addresses the points raised during the review process.

We look forward to receiving your revised manuscript.

Kind regards,

Frank T. Spradley

Academic Editor

PLOS ONE

Reviewers' comments:

Reviewer's Responses to Questions

**Comments to the Author**

1. Is the manuscript technically sound, and do the data support the conclusions?

Reviewer #1: Yes

2. Has the statistical analysis been performed appropriately and rigorously? 

Reviewer #1: Yes

3. Have the authors made all data underlying the findings in their manuscript fully available?

Reviewer #1: Yes

4. Is the manuscript presented in an intelligible fashion and written in standard English?

Reviewer #1: Yes

5. Review Comments to the Author

Reviewer #1: This is an interesting study that aims to substantiate the

association of enterolactone as a potential biomarker of gut health and CVD risk, we set out to examine

the association between urinary enterolactone concentration and blood pressure/risk of hypertension in a

large representative sample from the US population.

There are few comments to be considered in order to improve quality:

Line 139 is somehow confusing and same for lines 189_190 and suggest to rephrase.

Author's claim there are 2 novelties in the findings. Yet second novelty listed is a strength and not novelty

Did the authors consider antibiotic use by the participants?

Also if data for antihypertensive use was available; why didn't the authors add this to analysis to strengthen the design?

6. PLOS authors have the option to publish the peer review history of their article (what does this mean?). If published, this will include your full peer review and any attached files.

Reviewer #1: **Yes: **Fatme AlAnouti

---

## [Author Response · Author response to Decision Letter 0]

15 Feb 2024

Response to Reviewers

Reviewer #1: This is an interesting study that aims to substantiate the

association of enterolactone as a potential biomarker of gut health and CVD risk, we set out to examine the association between urinary enterolactone concentration and blood pressure/risk of hypertension in a large representative sample from the US population.

There are few comments to be considered in order to improve quality:

1. Line 139 is somehow confusing and same for lines 189_190 and suggest to rephrase. Author's claim there are 2 novelties in the findings. Yet second novelty listed is a strength and not novelty.

Author Response: We appreciate this comment and have clarified these two sentences as follows: Previous beginning line 139, now beginning line 145: “Individuals with hypertension were less likely to have a college education (22.1% vs. 25.4%; p=0.005), live at or below poverty (12.4% vs. 14.3%; p=0.01), be a current smoker (20.1% vs. 25.5%; p<0.001), have a body mass index in the ‘normal range’ of 18.6 to 24.9 (22.4% vs. 39.5%; p<0.001) and to have engaged in moderate physical activity in the past 30 days (33.4% vs. 36.9%; p<0.001).” 

The line that originally began on line 189/190 is now on lines 195-197, which we have clarified to point out that the novel finding is that even when accounting for all of these covariates, the strength of the association between enterolactone and hypertension was increased, rather than being attenuated, as one might expect. The line now reads as follows: “…2) accounting for age, sex, race, ethnicity, marital status, education, smoking status, and BMI strengthened, rather than attenuated, the association between enterolactone and hypertension, as well as the relationship between enterolactone and blood pressure.”

2. Did the authors consider antibiotic use by the participants?

Author Response: Thank you for this question, it is a really good point. We have revised our analysis to include an indicator variable in our models for self-reported use of antibiotics. The inclusion of this variable did not change our results in any meaningful way, but we believe that it is a more thorough analysis with the variable included. All tables have been updated accordingly.

3. Also if data for antihypertensive use was available; why didn't the authors add this to analysis to strengthen the design?

Author Response: We did have access to these data, and they were included in the analysis as part of the definition of our hypertension outcome variable. In the Methods lines 94-98 we described the definition of hypertension in our study as follows: “Ascertainment of hypertension involved a composite approach based on any indication of hypertension from (1) blood pressure readings based on the current American Heart Association guidelines (Hypertension defined as systolic blood pressure ≥130 mmHg or diastolic blood pressure ≥ 80 mmHg) [14], (2) self-reported diagnosis of hypertension by a healthcare professional, or (3) self-reported use of blood pressure medication.” Thus, the use of antihypertensive medication is part of the definition of our hypertension outcome variable. We have also clarified language regarding this throughout, including lines 136-144 in the Results: “Of the individuals with hypertension (either systolic >= 130 mmHg or diastolic >= 80 mmHg or self-reported diagnosis by a healthcare provider or self-reported taking blood pressure medication), 76.1% (SE=0.8%) had measured hypertension at time of examination (systolic >= 130 mmHg or diastolic >= 80 mmHg), 58.4% (SE=1.0%) reported being told by a healthcare provider that they had high blood pressure, and 46.7% (SE=1.0%) reported taking blood pressure medication.”

---

## [Decision Letter · Decision Letter 1]

1 Apr 2024

Association of enterolactone with blood pressure and hypertension risk in NHANES

PONE-D-23-26257R1

Dear Dr. Kevin S. Heffernan,

We’re pleased to inform you that your manuscript has been judged scientifically suitable for publication and will be formally accepted for publication once it meets all outstanding technical requirements.

Kind regards,

Awatif Abid Al-Judaibi, PhD

Academic Editor

PLOS ONE

Reviewers' comments:

Reviewer's Responses to Questions

**Comments to the Author**

1. If the authors have adequately addressed your comments raised in a previous round of review and you feel that this manuscript is now acceptable for publication, you may indicate that here to bypass the “Comments to the Author” section, enter your conflict of interest statement in the “Confidential to Editor” section, and submit your "Accept" recommendation.

Reviewer #1: All comments have been addressed

2. Is the manuscript technically sound, and do the data support the conclusions?

Reviewer #1: Yes

3. Has the statistical analysis been performed appropriately and rigorously? 

Reviewer #1: Yes

4. Have the authors made all data underlying the findings in their manuscript fully available?

Reviewer #1: Yes

5. Is the manuscript presented in an intelligible fashion and written in standard English?

Reviewer #1: Yes

6. Review Comments to the Author

Reviewer #1: Comments addressed and feel the manuscript is now suitable for publication. This an interesting research paper that will add to the scientific literature. Authors had clarified the confusing sentences highlighted and included data regarding use if antibiotics in analysis.

7. PLOS authors have the option to publish the peer review history of their article (what does this mean?). If published, this will include your full peer review and any attached files.

Reviewer #1: No

---

## [Editor Report · Acceptance letter]

2 May 2024

PONE-D-23-26257R1 

PLOS ONE

Dear Dr. Heffernan, 

I'm pleased to inform you that your manuscript has been deemed suitable for publication in PLOS ONE. Congratulations! Your manuscript is now being handed over to our production team.

Kind regards, 

on behalf of

Professor Awatif Abid Al-Judaibi 

Academic Editor

PLOS ONE